# Analysis of Korea's Green Technology Policy and Investment Trends for the Realization of Carbon Neutrality: Focusing on CCUS Technology

**Seok-ho Jung [1], Hongjin Kim [2], Youngjun Kang [3] and Eunhoo Jeong [4],***

1   Department of Earth and Environmental Sciences, Korea University, Seoul 02841, Korea; refresh@korea.ac.kr
2   Coalition for Our Common Future, 45 Supyo-ro, Jung-gu, Seoul 04551, Korea; coalition4@ourfuture.kr
3   Jeju Research Institute, 253, Ayeon-ro, Jeju 63147, Korea; yjkang@jri.re.kr
4   Department of Environmental Engineering, Seoul National University of Science &Technology (SeoulTech), 232 Gongneung-ro, Nowon-gu, Seoul 01811, Korea
*   Correspondence: jeongeunhoo@gmail.com; Tel.: +82-2-970-6623

**Abstract:** In 2008, the Republic of Korea announced the Low Carbon Green Growth vision as the national growth engine. This Green Growth vision invested in developing state-of-the-art green technologies to minimize greenhouse gas and pollutant emissions. Following a change in administration, Korean green technologies were re-categorized into six core technologies for climate change response in 2014, and ten core technologies for carbon-neutrality in 2021. The government proposed the realization of an inclusive green state in the 3rd Five-Year Green Growth Plan announced in 2019. Following the Green New Deal announced in 2020, green technology policies and investments continue, with the declaration of 2050 carbon neutrality. In the past two years, government policies from the 2050 Carbon Neutrality Strategy to the 2050 Carbon-Neutral Scenario consider Carbon Capture Utilization and Storage (CCUS) as an important technology to reduce $CO_2$ and meet carbon-neutral goals. This study examines major green technology policy trends, focusing on CCUS, beginning with the Green Growth era in 2008 to today. R&D investment costs related to green technology during the green growth period and CCUS-related investment costs over the past 10 years (2011–2020) are utilized in statistical analyses (correlation, trend) to investigate and analyze investment volatility in green and CCUS technologies related to climate change. Finally, the study will provide basic information for establishing CCUS-related R&D policies, which will continue to increase in achieving carbon neutrality.

**Keywords:** net-zero; CCUS; CCS; CCU; green technology

## 1. Introduction

Between 2008–2012, Low Carbon Green Growth (Low Carbon Green Growth, originally a vision, was the flagship policy during the Lee Myung-Bak administration) was the great engine powering the Republic of Korea. Guided by the National Strategy for Green Growth, infrastructure projects and investments were made in green technologies specializing in greenhouse gas emissions and pollution reductions. The president of Korea has a five-year term. Each regime presents a national policy management stance. From 2008 to 2012, green growth was the basis of national policy, and it was an important period that became the basis of the current carbon neutrality period. In the following two administrations, Korea's green technologies were reshuffled, first in 2014, into six core climate change response technologies, and again in 2021, into ten core carbon-neutral technologies. National-level investments in green technologies continued over the years, through the 3rd 5-year Green Growth Plan (2019)—which proposes a roadmap for constructing a Socially-inclusive Green Nation, and through to the 2020 Green New Deal and the 2050 Carbon Neutral policy [1–7].

In September 2021, maintaining the effort to meet the carbon-neutral goal, the Korean government enacted the *Framework Act on Carbon Neutrality and Green Growth*, scheduled to come into effect in 2022.

"The Basic Act on Carbon Neutralization aims to prevent severe effects of the climate crisis, by strengthening measures to reduce greenhouse gases and climate crisis adaptation, by resolving economic, environmental, and social inequalities that may occur in the process of transitioning to a carbon-neutral society, by developing, promoting, and vitalizing green technologies and green industries, improving the quality of life of the current and future generations, and protecting the ecosystem and climate system, and contributing to the sustainable development of the international community [7–9]."

Even in different administrations, green technologies remain within each administration's R&D technology policy and law. Carbon Capture Utilization and Storage (CCUS) technology is acknowledged as an essential carbon emission reduction technology by international organizations such as IEA, IPCC, and national policies like the 2050 Carbon-Neutral Scenario. The CCUS technology covered in this paper includes Carbon Capture and Storage (CCS) technology, which captures and stores carbon through dry scrubbing, wet scrubbing, and separation membrane and Carbon Capture Utilization (CCU) technology, transforming carbon into highly valued material.

The National Energy Technology Laboratory (NETL) under the U.S. Department of Energy (DOE) pioneered CCUS research, expanding from the traditional CCS to CCU [10].

CCUS technology for carbon reduction is being proposed in long-term low greenhouse gas emission development strategies (LEDS) of major countries.

It mainly suggested the use of technology in the energy and industrial sectors such as electricity and hydrogen. Germany mentioned CCUS technology as a strategy for the industrial sector. In the United States, CCUS technology was proposed as a major technical means of reducing carbon in the electricity sector. Japan proposed CCUS technology in the energy and industry sectors. The UK proposed the use of CCUS technology in the development of biomass and scenarios in which the hydrogen economy was activated [11].

In 2009, the Korean government included CCS in the twenty-seven green technologies for sustainable economic growth. In 2012, 127 experts for engineering or technology management assessed the economic and environmental feasibility of the twenty-seven technologies. Thirty-two of them assessed CCS as a technology that can balance economic development and environmental protection [2,4].

Recently in 2020, the Korean government proposed various policies relating to carbon neutrality, including the 2050 Carbon Neutrality Strategy, the 2021 Carbon Neutrality Implementation Plan, and the Carbon Capture and Utilization (CCU) Technology Innovation Roadmap. CCUS is evaluated as an important technology that can reduce $CO_2$ emissions. The 2050 Carbon-Neutral Scenario published in October 2021 shows two proposals targeting a carbon emission of 0. CCUS is set as the main absorption and removal method, along with forests and Direct Air Capture (DAC). In the carbon-neutral scenario, CCUS is suggested as the most important absorption and removal method with a maximum processing capacity of 85.2 million tons of $CO_2$-eq. Based on the National Greenhouse Gas Reduction Goals (NDC) announced in 2018, there was no carbon throughput using CCUS; but for 2050, the reduction amount of Plan A is calculated at 55.1 million tons $CO_2$-eq, while the reduction amount of Plan B was at 84.6 million tons $CO_2$-eq. If plan A is implemented, thermal power generation will be completely stopped, and if plan B is implemented, CCUS will be used and thermal power generation will be continued. Plan B has a 29.5 million ton $CO_2$-eq more reduction target using CCUS than Plan A and is highly dependent on technology [5,6,12–16].

In this study, major green technology policy trends from the Green Growth period (2008–2012) to the Carbon-Neutral period are reviewed, with a focus on CCUS. Additionally, R&D investment costs related to green technology during the green growth period and CCUS-related investment costs over the past 10 years (2011–2020) are utilized in statistical

analyses (correlation, trend) to investigate and analyze investment volatility in green and CCUS technologies related to climate change. Finally, the study serves as a basic overview for establishing CCUS-related R&D policies, the roles of which will continue to increase in achieving carbon neutrality.

## 2. Methodology

### 2.1. Literature Review and Data Collection

This research revised and developed information from the following literature: Research on Investment and Policy Directors for Climate Change Adaptation and the Green Technology Concept and Policy Development Direction Report by the Korea Institute of Science and Technology Evaluation and Planning, Analysis of the State of the Art of International Policies and Projects on CCU for Climate Change Mitigation with a Focus on the Cases in Korea by the Sustainability Journal, and the CCU Technology Innovation Roadmap by the Korean government, and assessed literature on Korea's carbon-neutrality policies and investment trends [2,4,10,15]. Academic DBs such as the National Digital Science Library (NDSL), DBpia, and the Korean Studies Information Service System (KISS), and websites of Korean government ministries such as that of the Ministry of Science and ICT (MSIT) were used to study literature on the carbon-neutral policy.

Table 1 lists the literature on Korea's major policies related to green technology, carbon neutrality, and CCUS from 2008 to 2021 [1–18].

**Table 1.** Korea's major policies related to green technology, carbon neutrality, and Carbon Capture Utilization and Storage (CCUS).

| Category | Title (Year) | Note |
|---|---|---|
| Green Technology | Declaration of the Low Carbon Green Growth Vision (2008) | National Liberation Day Speech |
| | Comprehensive Measures for R&D on Green Technology (2009) | National Science and Technology Committee |
| | Focused Green Technology Development and Commercialization Strategy (2009) | Green Growth Committee |
| | Five-Year Green Growth Plan (2009) | Green Growth Committee |
| | 10 Core Green Technologies (2010) | Green Growth Committee |
| | Inspection of 27 key green technology roadmaps (2011) | Green Growth Committee |
| | Creative Economy Realization Plan (2013) | Government ministries |
| | 2012 National R&D Project Survey and Analysis Report (2013) | GTC |
| | Roadmap for achieving national greenhouse gas reduction goals (2014) | Green Growth Committee |
| | The 2nd Five-Year Green Growth Plan (2014) | Green Growth Committee |
| | Strategy for Developing Core Technologies to Respond to Climate Change (2014) | Ministry of Science, ICT and Future Planning |
| | The 3rd Five-Year Green Growth Plan (2019) | Green Growth Committee |
| Carbon-neutrality and CCUS | The 2nd Science and Technology Strategy Meeting (2016) | Government |
| | National Strategic Project Demonstration Roadmap for Carbon Resourceization (2016) | Ministry of Science, ICT and Future Planning |
| | The Korean New Deal (2020) | Government ministries |
| | 2050 Carbon-Neutral Promotion Strategy (2020) | Ministry of Environment |
| | 2021 Carbon-Neutral Implementation Plan (2021) | Government ministries |
| | CCU Technology Innovation Roadmap (2021) | Government ministries |
| | 2050 Carbon-Neutral Scenario (2021) | Carbon Neutral Committee |
| | 2030 NDC (2021) | Carbon Neutral Committee |

Literature used for R&D investments included the *National R&D Program Investigation/Analysis* and the *R&D Activity Survey Report* by the Ministry of Science and ICT, science and technology statistics by the NTIS, *National R&D Project Survey and Analysis Report on Green Technology* by the Green Technology Center, and data from the *CCU Technology Innovation Roadmap*. Figure 1 illustrates the research process [4,15,18–20].

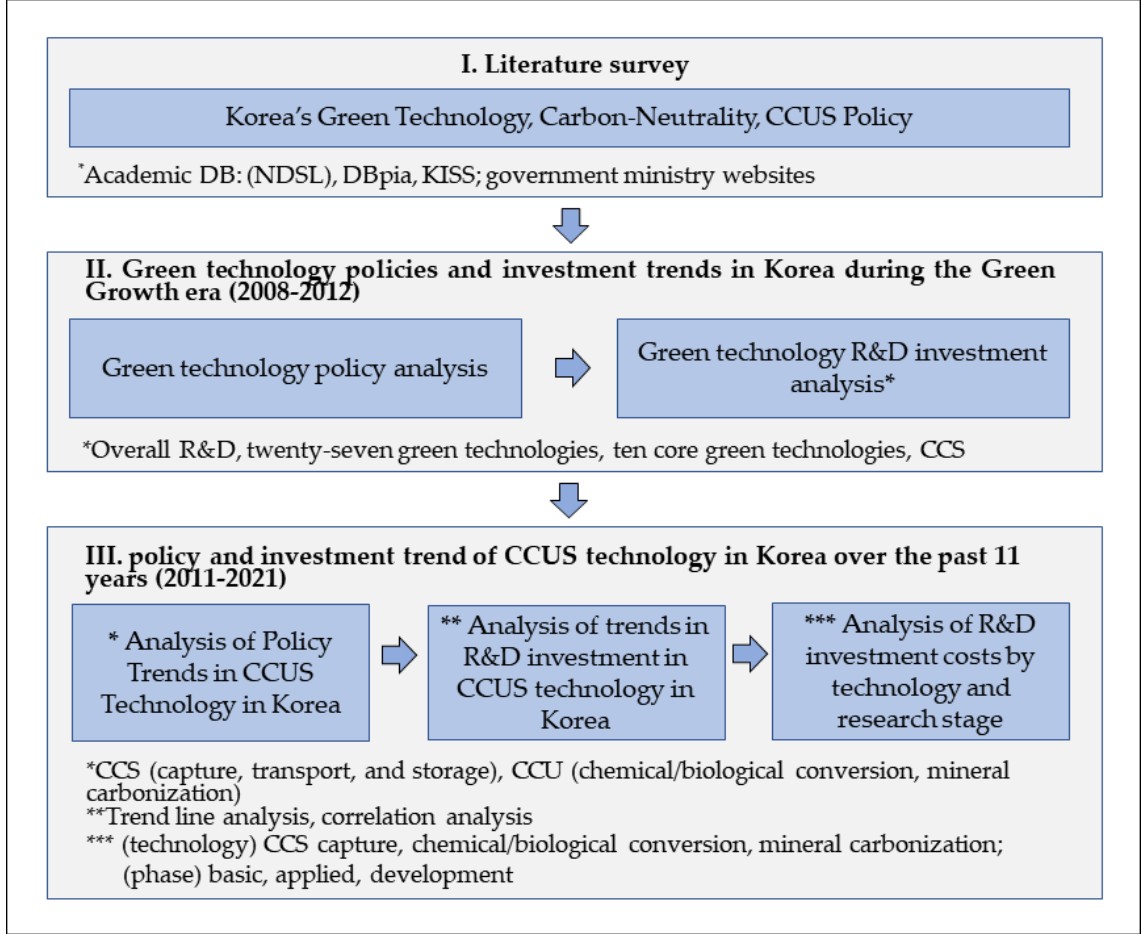

**Figure 1.** Research Process.

*2.2. Data Analysis*

Korea's national R&D investment in green technologies and carbon neutrality, including CCUS, in the period between 2008 to 2020, was statistically analyzed using Excel 206 and SPSS25.

**3. Green Technology Policy and Investment in the Green Growth Period**

The Framework Act on Carbon Neutralization in Korea, scheduled to take effect in 2022, emphasizes the importance of green technology for a carbon-neutral society. In the Framework Act on Low Carbon Green Growth, which took effect in 2010, green growth is defined as the balanced growth of the economy and the environment using green technology: technology that minimizes the emission of greenhouse gases and pollutants. Greenhouse gas reduction technology, energy use efficiency technology, clean production technology, clean energy technology, resource circulation, and eco-friendly technology fall within the scope of green technology (Basic Act on Low Carbon Green Growth, Carbon Neutral Act) [8,9].

In the 2008 National Liberation Day speech, the president of South Korea announced green growth as the National Development Paradigm. From 2008 to 2012, green growth promoted investment in green technology R&D as a new national growth engine; R&D

investment in green technology totaled about 1.46 trillion KRW in 2008 and expanded to about 2.71 trillion KRW in 2012. R&D investment in the CCS sector totaled 26.3 billion KRW in 2008 and expanded to about 74.2 billion KRW in 2012. Compared to the 1.9 times average increase in green technology R&D investment as a whole, CCS R&D investment increased 2.8 times, showing a higher investment growth rate. Figure 2 shows the total amount of green technology R&D investment during the Green Growth era, including CCS [2,18,19].

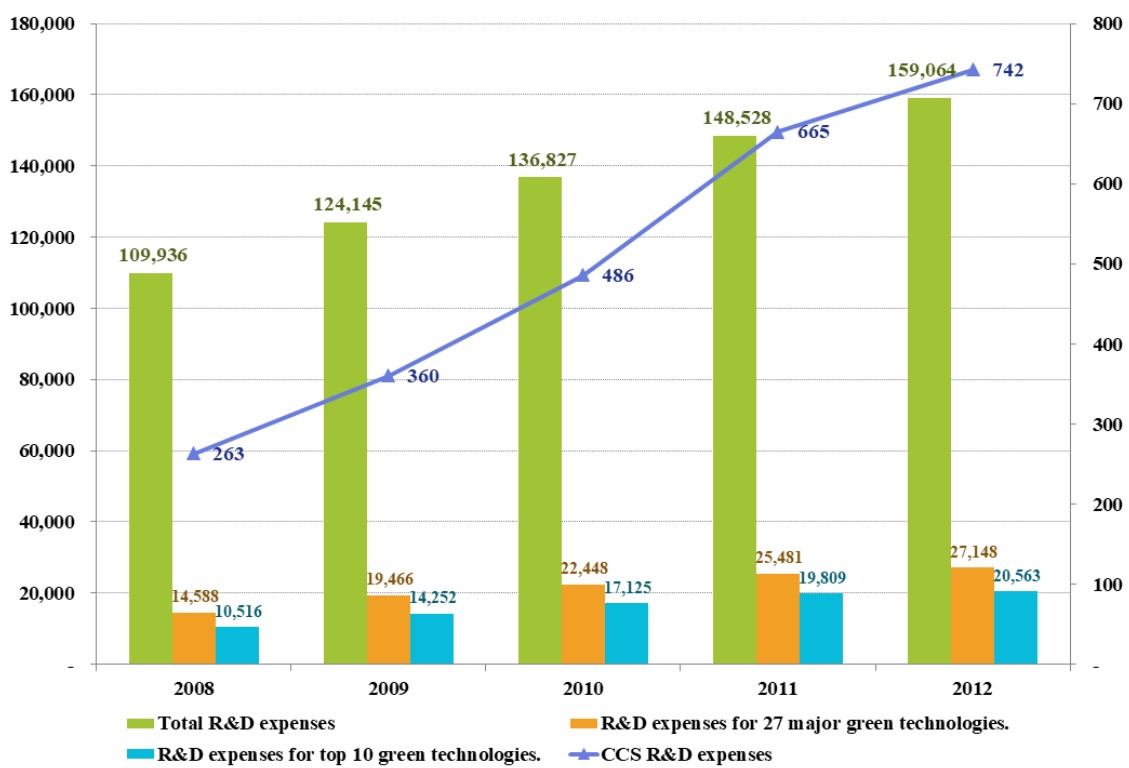

**Figure 2.** Green Technology Investment (2008–2012) (data from Green Technology Center, 2013).

Twenty-seven key green technologies were presented in the Five-Year Green Growth Plan announced by the Korean government in 2009. Ten core green technologies were further revealed in the seven tasks for green growth announced in 2010, along with a plan to promote new growth engines. The ten core green technologies include secondary batteries, future nuclear power, advanced water treatment, CCS, smart grids, LED lighting displays, green IT, solar cells, green cars, and fuel cells. Figure 3 shows the investment trends of 27 major green technologies from 2008 to 2012 [2,3,18].

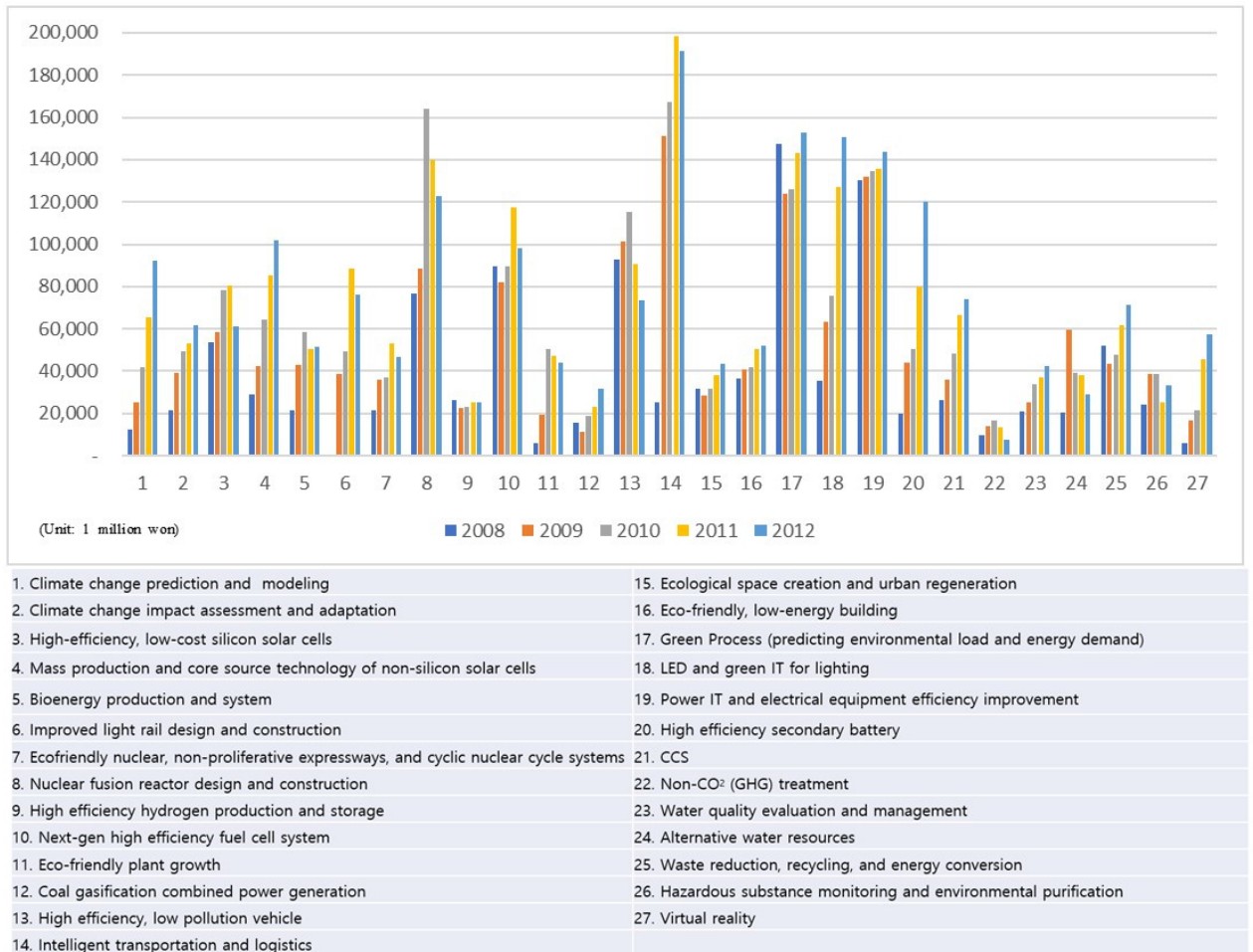

**Figure 3.** Investment trend of 27 Green Technologies from 2008 to 2012 (data from Green Technology Center, 2013).

## 4. Korean CCUS Policy and R&D Investment in the Past Eleven Years (2011–2021)

### 4.1. CCUS Policy

In 2014, MSIT, preceding the Ministry of Science and ICT, announced six core technologies in three major fields through the *Climate Change Response Core Technology Development Strategy*: solar cell, fuel cell, and bio cell technologies were categorized under fossil fuel replacement; secondary battery and power IT (EMS) under greenhouse gas emission reduction; and CCS technology under greenhouse gas treatment. In 2012, the ministry announced the increase of R&D investment from 2.2 trillion KRW in climate change response technology and 0.77 trillion KRW in six core technologies by more than 50% each by 2020 [4].

Greenhouse gas emissions reduction efforts from government ministries such as MSIT, MOTIE (Ministry of Trade, Industry, and Energy), MOF (Ministry of Oceans and Fisheries), and ME (Ministry of Environment), have been reflected in their investment plans for CCS. In the Climate Change Response Technology Development Project, for instance, MSIT supported securing leading source technologies with large greenhouse gas reduction potentials to cope with the climate crisis. In June 2011, the Korea CCS 2020 Basic Plan was published and the Korea Carbon Capture and Sequestration R&D Center (KCRC) was established in the same year to promote the 172.7 billion KRW national project in the period between 2011 and 2019. MOTIE strategized the Greenhouse Gas Treatment Technology Development Project centered on large-scale CCS demonstration and commercialization projects worth 93.5 billion KRW. MOF planned the 90 billion KRW $CO_2$ Marine Treatment Technology Development Project, while the ME planned the 20 billion KRW $CO_2$ Storage

Environment Management Technology Development Project. Finally, the Korea $CO_2$ Storage Environmental Management Research Center (K-COSEM) was formed in 2014 as part of a multi-ministry effort by the ME, MSIT, MOTIE, and MOF [4].

In 2016, Korea designated CCU as one of the nine major national strategic projects during the Second Science and Technology Strategy Meeting. Starting from this, the notion of CCS—mainly concerned with reducing and treating existing greenhouse gases—was expanded to CCUS, including the concept of CCU. In terms of policy, the National Strategic Project Demonstration Roadmap for Carbon Resourceization was announced in 2016; in 2017, a national strategic research center worth 49.5 billion KRW was established with the combined effort from MOLI, MOTIE, and the European Union (EU) [10].

In late 2018, the EU announced its carbon neutrality goal by 2050 through the mammoth Green Deal. By the end of 2020, 128 countries declared carbon neutrality targets. In July 2020, Korea announced its effort towards reaching carbon neutrality and constructing the circular economy through the Green New Deal, a subset of the Korean New Deal. The carbon neutrality target became official in October 2020 at the National Assembly.

Until recently, the government announced the 2050 Carbon-Neutral Promotion Strategy, which contains the 2021 Carbon Neutral Implementation Plan, CCU Technology Innovation Roadmap, and the 2050 Carbon-Neutral Scenario as detailed plans to meet the carbon neutrality goal [6,12–16].

The CCU Technology Innovation Roadmap indicates milestones up to 2050 of the fifty-nine detailed technologies in the five major technology areas, including $CO_2$ capture, chemical conversion, biological conversion, mineral carbonization, and other forms of carbon utilization. This is meaningful as it presents milestones and KPIs for each detailed key technology and the need to commercialize CCUS technology to realize carbon neutrality. Figure 4 shows the classification system of CCU technology of the roadmap configured in the form of Work Breakdown Structure (WBS) [15].

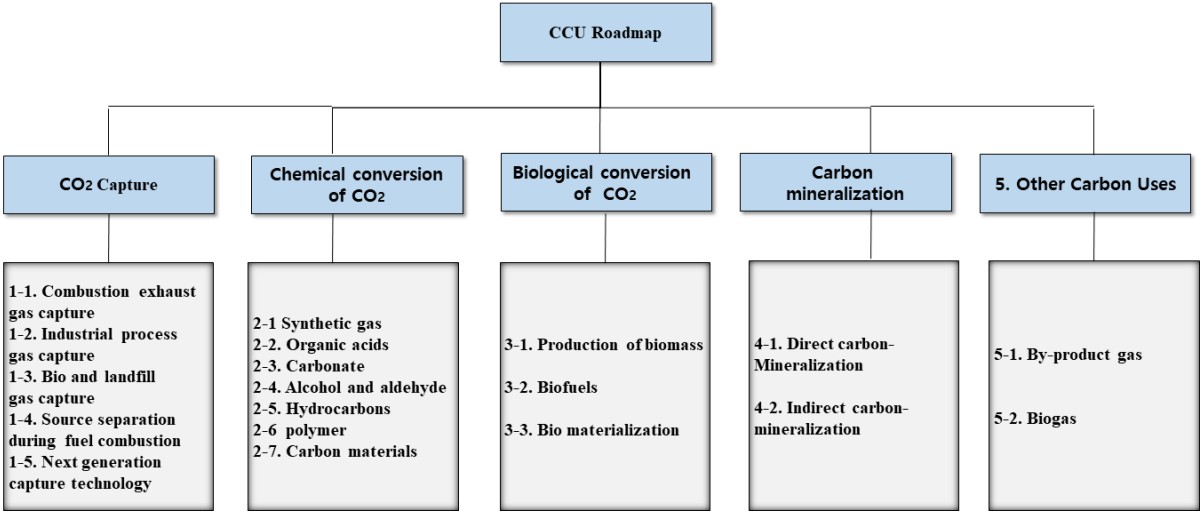

**Figure 4.** Carbon Capture Utilization (CCU) Technology Innovation Roadmap (data from MSIT, 2021).

Policy-wise, Korea's CCUS technology expanded from CCS to CCUS—the latter including CCU. CCS was first proposed as one of the twenty-seven green technologies during the 2009 Green Growth era, and one of the six core technologies in responding to climate change in the 2014 Creative Economy era. In 2021, through the *Carbon Neutral Technology Innovation Promotion Strategy*, it is considered one of the ten core technologies—waiting for commercialization—supporting the industry's transition to low-carbon. Figure 5 lists core technologies for each major technology policy of the Korean government [5].

| Ten core carbon-neutral technologies (2021~) |
| --- |
| 1. Ultra-efficient and large-scale solar/wind power |
| 2. Securing all technologies in the hydrogen cycle |
| 3. Securing leading technologies in bioenergy |
| 4. Steel and cement industry's transition to low carbon |
| 5. Low-carbon next-gen petrochemical industry |
| 6. Maximizing industrial process efficiency |
| 7. Carbon-free next-gen transportation technology |
| 8. Carbon-neutral building |
| 9. Digitalized optimization efficiency |
| 10. Securing technology for CCUS commercialization |

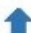

| Six core technologies for climate change response (2014~) | | |
| --- | --- | --- |
| (fossil fuel replacement) | (GHG treatment) | (GHG reduction) |
| 1. Solar cell 2. Fuel cell 3. Bio cell | 4.CCS | 5. Secondary battery 6.Power IT (EMS) |

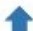

| Twenty-seven Green Technologies (2009~) | |
| --- | --- |
| 1. Climate change prediction and modeling | 15. Ecological space creation and urban regeneration |
| 2. Climate change impact assessment and adaptation | 16. Eco-friendly, low-energy building |
| 3. High-efficiency, low-cost silicon solar cells | 17. Green Process (predicting environmental load and energy demand) |
| 4. Mass production and core source technology of non-silicon solar cells | 18. LED and green IT for lighting |
| 5. Bioenergy production and system | 19. Power IT and electrical equipment efficiency improvement |
| 6. Improved light rail design and construction | 20. High-efficiency secondary battery |
| 7. Eco-friendly nuclear, non-proliferative expressways, and cyclic nuclear cycle systems | 21. CCS |
| 8. Nuclear fusion reactor design and construction | 22. Non-CO$_2$ (GHG) treatment |
| 9. High-efficiency hydrogen production and storage | 23. Water quality evaluation and management |
| 10. Next-gen high-efficiency fuel cell system | 24. Alternative water resources |
| 11. Eco-friendly plant growth | 25. Waste reduction, recycling, and energy conversion |
| 12. Coal gasification combined power generation | 26. Hazardous substance monitoring and environmental purification |
| 13. High efficiency, low pollution vehicle | 27. Virtual reality |
| 14. Intelligent transportation and logistics | |

**Figure 5.** Core technologies for each major technology policy of the Korean government. (data from the Korea Institute of Science and Technology Evaluation and Planning (KISTEP), 2012; KISTEP, 2014; MSIT, 2021).

*4.2. CCUS R&D Investment (2011–2020)*

Figure 6 illustrates R&D investment in CCUS over the past ten years (2011 to 2020) based on the CCU Technology Innovation Roadmap data. It is classified into three categories: capture, utilization, and transportation and storage. Over the past ten years, R&D

investment in CCUS generally decreased to about 469,243 million KRW, paralleling similar decreases in investments in capture and transportation. The sharp decline in budget for transportation storage since 2017 is attributed to the earthquake near Yeongil Bay in Pohang and safety reviews. Expert reviews indicate that the possibility of an earthquake in Yeongil Bay is very low.

On the other hand, R&D investment in the utilization sector has been on the rise since 2016, indicating that investments expanded from a CCS-centered portfolio to one of CCUS that includes CCU [15,20].

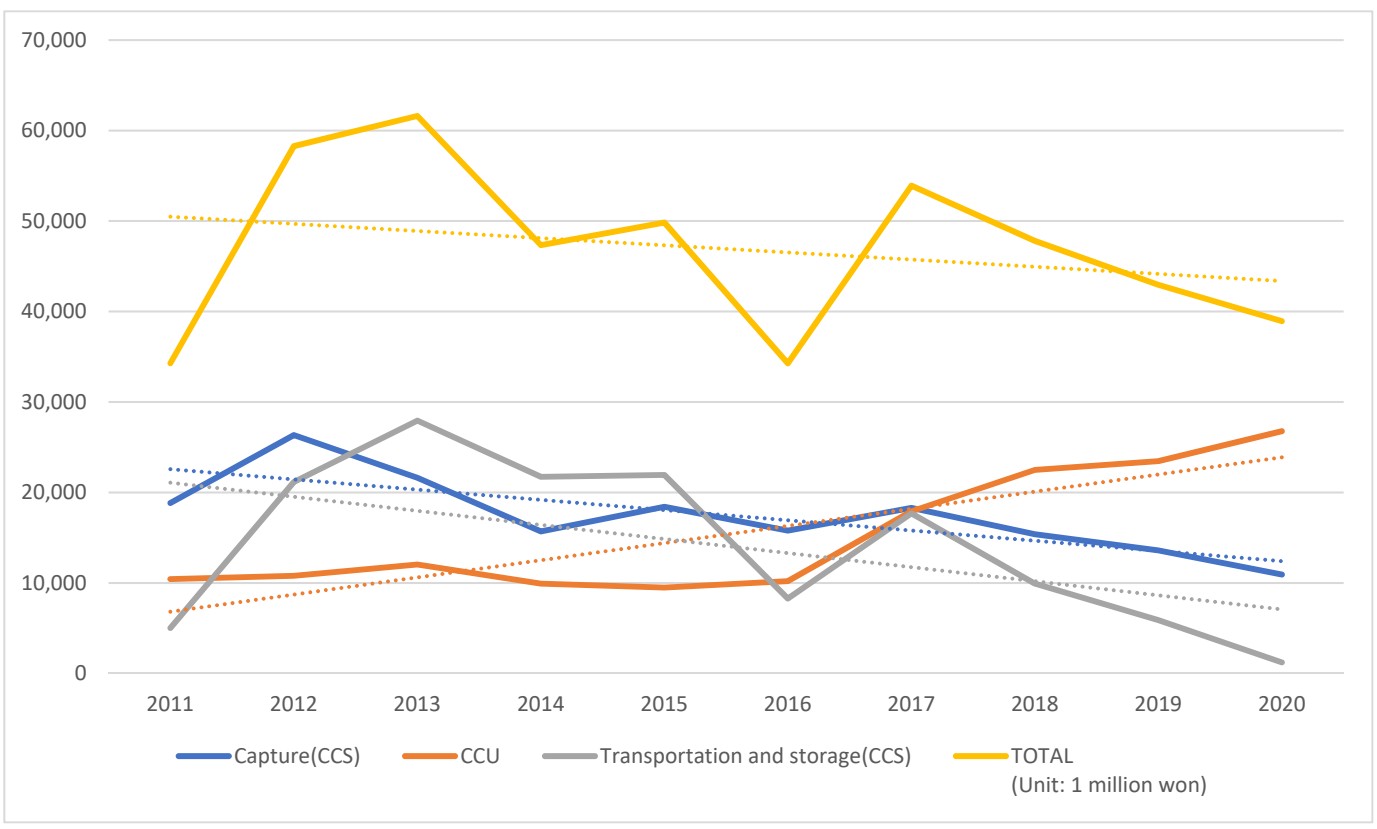

**Figure 6.** R&D investment in CCUS over the past ten years from 2011 to 2020 (data from MSIT, 2021).

Table 2 shows the result of the correlation between R&D investment in CCUS and capture, utilization, and transportation/storage. The capture ($p \leq 0.05$) and transportation/storage ($p \leq 0.01$) have significant correlations with volatility in CCUS investments.

**Table 2.** The result of the correlation between R&D investment in CCUS.

|  | Capture | Utilization | Transportation/Storage | Total |
|---|---|---|---|---|
| Pearson correlation | 0.653 * | −0.162 | 0.849 ** | 1 |
| *p*-value | 0.041 | 0.655 | 0.002 | |
| N | 10 | 10 | 10 | 10 |

*. Correlation significant at 0.05-level (both sides), **. Correlation significant at 0.01-level (both sides).

Figure 7 shows R&D investment in CCU from 2011 to 2020 based on data retrieved from the CCU Technology Innovation Roadmap. It is classified into three categories: $CO_2$ chemical conversion, $CO_2$ bioconversion, and mineral carbonization. Over the past ten years, R&D investment in CCU has risen to about 153,536 million KRW. $CO_2$ chemical conversion, $CO_2$ bioconversion, and mineral carbonization technology show increasing trends.

Since 2016, R&D investment in chemical conversion and mineral carbonization has increased rapidly, reflecting the project cost of the National Strategic Project for Carbon Re-

sourceization. In addition, R&D investment in next-generation carbon resource conversion projects, such as the Korea CCS 2020 project and the C1 Refineries project, is reflected in the increase in chemical conversion investments [10,15].

CCS technology investment costs showed a decreasing trend. CCU technology investment costs showed an increasing trend. There was no significant change in the CCUS overall technology investment costs, including CCS and CCU.

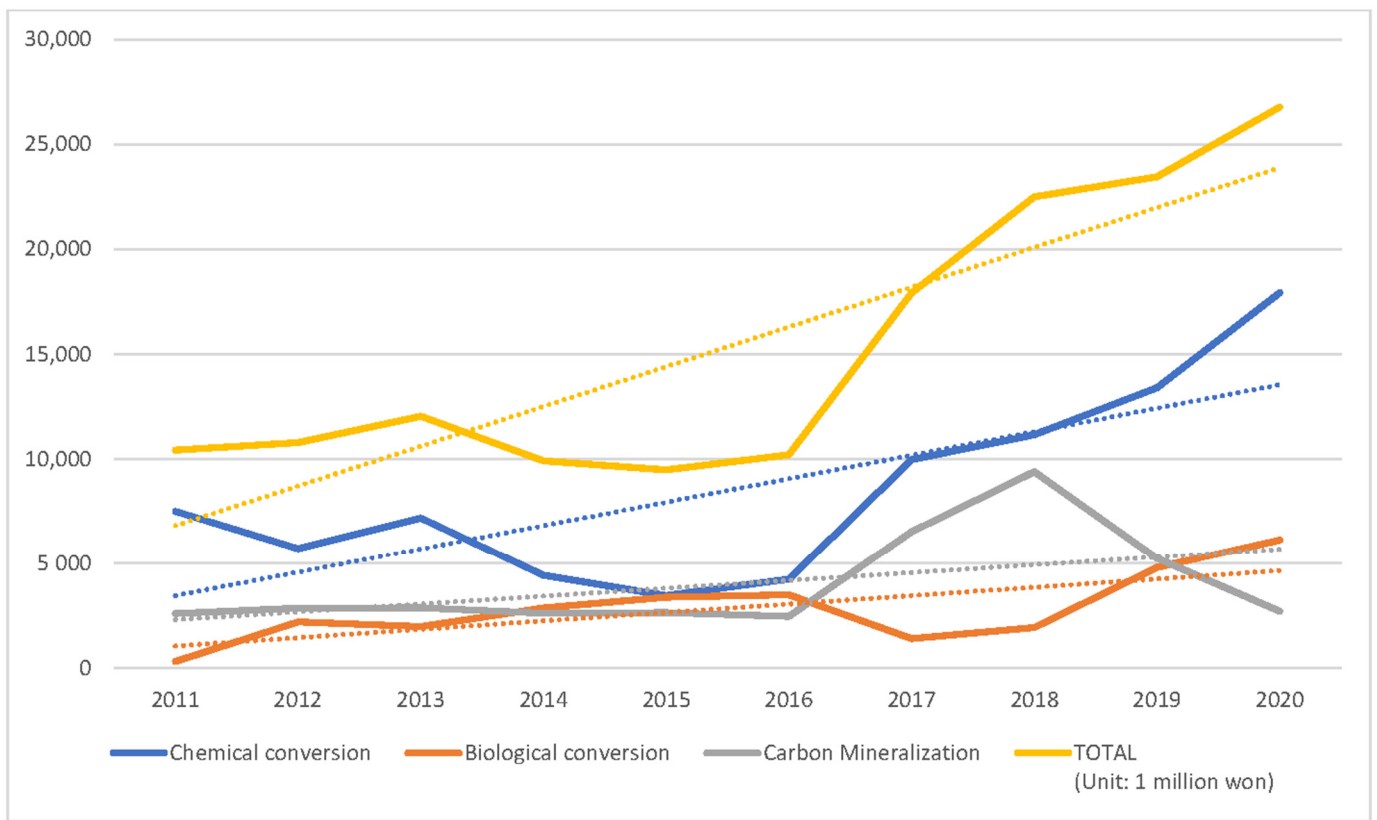

**Figure 7.** R&D investment in CCU over the past ten years from 2011 to 2020 (data from MSIT, 2021).

Correlation analysis between R&D investment in CCU over the past ten years and (i) chemical conversion, (ii) biological conversion, and (iii) carbon mineralization are shown in Table 3 below. Only $CO_2$ chemical conversion shows a significant correlation to CCU investment with volatility.

**Table 3.** The result of the correlation between R&D investment in CCU.

|  | Chemical Conversion | Biological Conversion | Carbon Mineralization | Total |
|---|---|---|---|---|
| Pearson correlation | 0.957 ** | 0.540 | 0.566 | 1 |
| *p*-value | 0.000 | 0.107 | 0.088 |  |
| N | 10 | 10 | 10 | 10 |

**. Correlation is significant at 0.01-level (both sides).

The Ministry of Science and ICT, the major R&D-related ministry in Korea, categorizes R&D into basic, applied, and development research. Basic research is defined as experimental results or theoretical studies mainly conducted to acquire new scientific knowledge underlying natural phenomena and observable objects without targeting specific applications or uses.

Applied research is defined as an original study conducted to acquire new scientific knowledge for special and practical purposes and goals. Development research is defined

as a systematic activity to substantially improve what has already been produced or installed, such as the production of new materials/products and devices, installation of new processes/systems or services, and knowledge acquired by research and experimental experience [21].

Figure 8 analyzes R&D investment for each of the research stages in major CCUS sub-sections (CCS capture, $CO_2$ chemical conversion, $CO_2$ bioconversion, and Carbon mineralization). The research stage is divided into basic, applied, and development research. $CO_2$ capture and mineral carbonization make up a large proportion of development research compared to basic and applied research, indicating that research for commercialization in these two areas is being conducted earlier than other CCUS technologies.

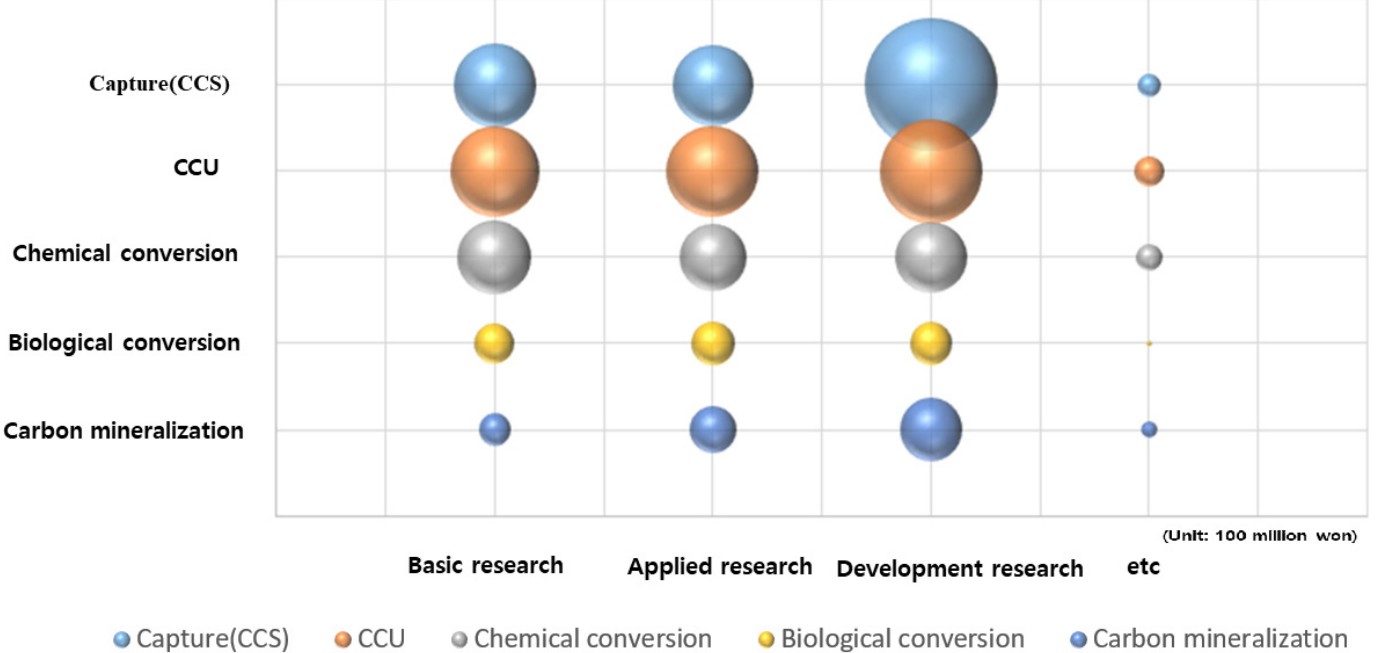

**Figure 8.** R&D investment for each of the research stages in major CCUS sub-sections in Korea from 2011 to 2020 (data from MSIT, 2021).

Major institutions around the world, including the National Academy of Sciences, Engineering, and Medicine (NASEM) (U.S) and the New Energy and Industrial Technology Development Organization (NEDO) (Japan) evaluate that, among various CCUS technologies, CCS and carbon mineralization are the two with the highest probability of commercialization. Under such consideration, the trend in CCUS R&D investment shows practical significance compared to the global trends.

The difference among basic, applied, and development research R&D in $CO_2$ chemical conversion and biological conversion investment is not significant [15,22,23].

## 5. Conclusions

As a new national growth engine, Korea's Low-Carbon Green Growth promoted the advancement of green technology as a whole. Korea's Framework Act on Carbon Neutralization, scheduled to take effect in 2022, also emphasizes the importance of green technology for a carbon-neutral society. Korea's green technology has been continuously modified and developed depending on the policy and technical environment, resulting in twenty-seven green technologies in 2009, six core technologies to respond to climate change in 2014, and ten core technologies in carbon neutrality in 2021. Beginning with the designation of CCU as one of the nine national strategic projects at the Second Science and Technology Strategy Meeting, this GHG emissions reduction technology incorporated CCU; afterward being noted as the more comprehensive CCUS. This is significant in that

NETL pioneered the research by expanding the scope of the technology from CCS to CCUS in early 2010.

A host of countries followed the EU's Green Deal and carbon neutrality goal announcement. Korea, too, announced its Green New Deal, 2050 Carbon Neutral Promotion Strategy, 2021 Carbon Neutral Implementation Plan, CCU Technology Innovation Roadmap, and 2050 Carbon-Neutral Scenario.

In Korea's carbon neutrality policy, CCUS' contribution to greenhouse gas reduction continues to increase. In the carbon-neutral scenario finalized and announced in October 2021, CCUS comprised the most important absorption and removal plan, with a maximum processing capacity of 85.2 million tons of $CO_2$-eq. In 2050, the net emission target was 55.1 million tons of $CO_2$-eq in Plan A and 84.6 million tons of $CO_2$-eq in Plan B. Compared to Plan A, which completely halts thermal power generation, Plan B continues the operation of thermal power generation. It uses CCUS to reduce 29.5 million more tons of $CO_2$-eq than Plan A.

Between 2011 and 2020, R&D investment in CCUS was about 469,243 million KRW, showing a gradual decline. Similarly, investment in CCS capture and CCS transportation storage are on the decline. On the other hand, R&D investment in the utilization sector has been on the rise since 2016, indicating an expanded coverage from CCS-centered R&D support policies to CCUS technologies including CCU. CCUS R&D investment shows a statistically significant correlation between CCS capture and CCS transportation storage.

In the same period, R&D investment in CCU has risen to about 153,536 million KRW. $CO_2$ chemical conversion, $CO_2$ bioconversion, and mineral carbonization investments are all on the rise. The significant increase in chemical conversion and mineral carbonization since 2016 can be attributed to the National Strategic Project. CCU R&D investment shows a high correlation with $CO_2$ chemical conversion.

In the analysis of basic, applied, and development R&D budgets for CCS $CO_2$ capture, $CO_2$ chemical conversion, $CO_2$ biological conversion, and mineral carbonization, CCS $CO_2$ capture and mineral carbonization have particularly large shares in development research compared to others. In comparison with the global trend, this correlation is judged to show practical significance.

This study analyzed major policies and investment trends related to green technology, focusing on CCUS since the Green Growth era in 2008 to 2021. Given that, over the years the weight of CCUS has increased to meet the national GHG reduction goals, this study endeavors to contribute to the literature on CCUS to meet carbon-neutral goals and serve as a basic foundation for the implementation of the CCU technology innovation roadmap and carbon-neutral scenarios.

**Author Contributions:** S.-h.J.: Writing–Original draft, Analysis, Investigation, Review & Editing; H.K.: Writing, Funding Acquisition; Y.K.: Analysis, Review & Editing, Supervision; E.J.: Analysis, Writing, Investigation, Supervision, Funding Acquisition. All authors have read and agreed to the published version of the manuscript.

**Funding:** This work was supported by a "Convergence Research Policy Fellowship" from the MSIT and KIST Convergence Research Policy Center And "Coalition for Our Common Future".

**Institutional Review Board Statement:** Not applicable.

**Informed Consent Statement:** Not applicable.

**Data Availability Statement:** Not applicable.

**Conflicts of Interest:** The authors declare no conflict of interest.

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
