# Peer review of "Analysis of Korea’s Green Technology Policy and Investment Trends for the Realization of Carbon Neutrality: Focusing on CCUS Technology"

_processes, doi:10.3390/pr10030501_

Round 1

Reviewer 1 Report

This study summarizes the analysis of green technology policies and investment trends for achieving carbon neutrality in Korea with a focus on CCUS technology, but the article still needs significant revision in terms of in-depth analysis of policies and normative aspects, and it is suggested that the following aspects be added and revised.

  1. The study is divided into two periods, but the basis for the division into two periods from 2008 to 2021 is not given. In addition, why 2012 was chosen as the time point for the division, can it be reflected in the introduction?
  2. Table 1 in the second part of the article lists the relevant policies and literature of Korea from 2008-2021, and it is suggested that only the policies are shown in the table.
  3. It is suggested to add the innovation of this study and the difference with other studies.
  4. In Figure 1, why the starting year of Part III is 2011, which is not connected with 2012 in Part II, does this cause the same policy to appear in two different periods and affect the results?
  5. Figure 3 only lists the policies issued in the period 2008-2011, which are only more specific dates compared to Table 1, and there is no need to discuss them again. In addition, rows 146-153 show the specific 27 key green technologies and 10 core green technologies. It is suggested to add which technologies are involved in the relevant policies during 2008-2011 and the specific funding amount for each technology, and to supplement the figure for display.
  6. The content displayed in Figure 4 is also displayed in Table 1, and it is suggested that no duplicate display is made.
  7. Part 4.1 is titled CCUS policy, but this part only illustrates that CCUS is proposed as a core technology by the government, and lacks specific which policies involve CCUS technology, as well as the level of attention and changing trend of the government on CCUS technology, and it is suggested to supplement this part.
  8. As seen in Figure 7, there is no significant upward trend in investment in CCUS technology in Korea in the past decade, can a brief explanation of this phenomenon be provided.
  9. Capture, utilization, transportation and storage are all part of the CCUS technology chain. When Figure 7 is presented, it is sufficient for the legend to label the corresponding technology chain only, and abbreviations are not recommended, and this is also the case when related explanations are made thereafter.
  10. The figure name of Figure 9 should be placed at the bottom of the picture.

Author Response

1.The study is divided into two periods, but the basis for the division into two periods from 2008 to 2021 is not given. In addition, why 2012 was chosen as the time point for the division, can it be reflected in the introduction?

We are grateful to the reviewer for acknowledging the importance of our study and suggesting valuable comments.

We revised the next part. Line 37

2.Table 1 in the second part of the article lists the relevant policies and literature of Korea from 2008-2021, and it is suggested that only the policies are shown in the table.

It is suggested to add the innovation of this study and the difference with other studies.

We are grateful to the reviewer for acknowledging the importance of our study and suggesting valuable comments.

We revised the next part. Line 17, Line 37, Line70, Line 132

3.In Figure 1, why the starting year of Part III is 2011, which is not connected with 2012 in Part II, does this cause the same policy to appear in two different periods and affect the results?

We are grateful to the reviewer for acknowledging the importance of our study and suggesting valuable comments.

Part 2 is the period when Korea proposed green growth as a major policy from 2008 to 2012. The period from 2011 to 2020 suggested by Part 3 is when data on CCUS-related R&D investment costs can be secured. Due to data consistency and reliability issues, the period for data acquisition has been suggested since 2011.

4.Figure 3 only lists the policies issued in the period 2008-2011, which are only more specific dates compared to Table 1, and there is no need to discuss them again. In addition, rows 146-153 show the specific 27 key green technologies and 10 core green technologies. It is suggested to add which technologies are involved in the relevant policies during 2008-2011 and the specific funding amount for each technology, and to supplement the figure for display.

We are grateful to the reviewer for acknowledging the importance of our study and suggesting valuable comments.

We revised the next part. Line188

  1. The content displayed in Figure 4 is also displayed in Table 1, and it is suggested that no duplicate display is made.

We are grateful to the reviewer for acknowledging the importance of our study and suggesting valuable comments.

I reflected it in the manuscript.

6.Part 4.1 is titled CCUS policy, but this part only illustrates that CCUS is proposed as a core technology by the government, and lacks specific which policies involve CCUS technology, as well as the level of attention and changing trend of the government on CCUS technology, and it is suggested to supplement this part.

We are grateful to the reviewer for acknowledging the importance of our study and suggesting valuable comments.

We revised the next part. Line96, Line238 Line248,

  1. As seen in Figure 7, there is no significant upward trend in investment in CCUS technology in Korea in the past decade, can a brief explanation of this phenomenon be provided.

We are grateful to the reviewer for acknowledging the importance of our study and suggesting valuable comments.

We revised the next part. Line 282

  1. Capture, utilization, transportation and storage are all part of the CCUS technology chain. When Figure 7 is presented, it is sufficient for the legend to label the corresponding technology chain only, and abbreviations are not recommended, and this is also the case when related explanations are made thereafter.

We are grateful to the reviewer for acknowledging the importance of our study and suggesting valuable comments.

We revised the next part. Line258

9.The figure name of Figure 9 should be placed at the bottom of the picture.

We are grateful to the reviewer for acknowledging the importance of our study and suggesting valuable comments.

We revised the next part. Line304

Reviewer 2 Report

This topic of this manuscript is interesting and useful for the researchers focusing on the evaluation and development of the Low-Carbon Clean Technologies. However, the authors only review the Korea’s policies and research processes on carbon capture utilization and storage (CCUS) in the recent ten years, and do not give any depth analysis on the feasibility and economics of the available CCUS technologies. My comments are as following:

  1. The core of these policy trends lies on the intrinsic advance of the CCUS technologies; however, this manuscript lacks the discussion on the progress of the current CCUS technologies.
  2. What is the difference of the policy trends between Korea and other countries, e.g., U.S.A, P.R.C and Japan?
  3. The authors need to add a section to analysis the global policy or development trends of the CCUS technologies.
  4. The writing of the manuscript is very poor, and needs further improvement, e.g., the format of the text in this manuscript.
  5. The quality of the figures in this manuscript is also poor, and must be further improved. In addition, the format of the Tables is also confused.
  6. There are many abbreviations in this manuscript, which can be defined at the end of the manuscript.
  7. Give complete form of the data in Table 2, e.g., .653, -.162?
  8. What are the dot lines in Figures 7 and 8?
  9. Most of the references are Korea’s government reports, which is hard for English readers.

Author Response

1.The core of these policy trends lies on the intrinsic advance of the CCUS technologies; however, this manuscript lacks the discussion on the progress of the current CCUS technologies.

We are grateful to the reviewer for acknowledging the importance of our study and suggesting valuable comments.

We revised the next part. Line 70, Line 96, Line188, Line246

2.What is the difference of the policy trends between Korea and other countries, e.g., U.S.A, P.R.C and Japan?

We are grateful to the reviewer for acknowledging the importance of our study and suggesting valuable comments.

We revised the next part. Line 70

3.The authors need to add a section to analysis the global policy or development trends of the CCUS technologies.

We are grateful to the reviewer for acknowledging the importance of our study and suggesting valuable comments.

We revised the next part. Line 68, Line 70, Line 308

4.The writing of the manuscript is very poor, and needs further improvement, e.g., the format of the text in this manuscript.

We are grateful to the reviewer for acknowledging the importance of our study and suggesting valuable comments.

We revised and supplemented the manuscript.

5.The quality of the figures in this manuscript is also poor, and must be further improved. In addition, the format of the Tables is also confused.

There are many abbreviations in this manuscript, which can be defined at the end of the manuscript.

Give complete form of the data in Table 2, e.g., .653, -.162?

We are grateful to the reviewer for acknowledging the importance of our study and suggesting valuable comments.

We revised and supplemented the manuscript.
Line258, Line 268, Line 287

6.What are the dot lines in Figures 7 and 8?

We are grateful to the reviewer for acknowledging the importance of our study and suggesting valuable comments.

The dotted line is the trend line. The trend line is one of the statistical techniques. If the right side faces upward, it is on the rise. If the right side faces downward, it is on the downward trend.

  1. Most of the references are Korea’s government reports, which is hard for English readers.

We are grateful to the reviewer for acknowledging the importance of our study and suggesting valuable comments.

Unfortunately, the official report is the Korean version.

Round 2

Reviewer 1 Report

The overall revision of the article is not significant, but there are still major problems and it is recommended to continue to make corrections.

  1. The article has not been revised in terms of in-depth analysis of policies and the standardization of the article, and the conclusion is only a review of the current policy situation, lacking an in-depth analysis of the reasons for the current situation.
  2. The second part of the article collects relevant policies related to CCUS, but the analysis angle is too single, only sorting out the existing policies, lacking comparison between policies, suggesting different dimensions of analysis. For example, different policies may focus on standard setting, technology development, project demonstration, future deployment planning, etc.
  3. The analysis in the third part of the article is too homogeneous and is only a simple recapitulation of existing policies, lacking an overall evaluation of the relevant policies during the period and the directions that should be focused on in the future, which could lead to the content of policies for the next period.
  4. Section 4.1 is titled CCUS policies, but this section is mostly irrelevant, only stating that CCUS has been proposed by the government as a core technology, and lacks specific policies that address CCUS technology, as well as the level of government attention and trends in CCUS technology.
  5. Section 4.2 is a simple recapitulation of existing data and is too homogeneous in its analysis.
  6. The conclusion suggests adding a section on relevant policy insights, indicating the shortcomings of existing CCUS policies and the directions on which they should focus in the future.

Author Response

  • The article has not been revised in terms of in-depth analysis of policies and the standardization of the article, and the conclusion is only a review of the current policy situation, lacking an in-depth analysis of the reasons for the current situation.

We are grateful to the reviewer for acknowledging the importance of our study and suggesting valuable comments.

This study focuses on the linkage and trend analysis of CCUS-centered policies to reduce greenhouse gas emissions in Korea for a long period from 2008 to 2021. This study is a series of papers. In-depth analysis of CCUS policies, institutions, and investments is already included in the paper below prepared by the author. This data was used as a reference for the paper.

(Page16, Line 393)

2.The second part of the article collects relevant policies related to CCUS, but the analysis angle is too single, only sorting out the existing policies, lacking comparison between policies, suggesting different dimensions of analysis. For example, different policies may focus on standard setting, technology development, project demonstration, future deployment planning, etc.

We are grateful to the reviewer for acknowledging the importance of our study and suggesting valuable comments.

Korea's major R&D policies are established as basic plans, mid- to long-term plans, and annual implementation plans for each regime and field. In addition, detailed plans will be established and implemented as part of the implementation plan.

As such, there is a limit to standardization as there are differences in character and level for each plan. Therefore, it is more appropriate to see the connection of each plan. The Korean New Deal (2020), 2050 Carbon Neutral Promotion Strategy (2020), 2021 Carbon Neutral Implementation Plan (2021), CCU Technology Innovation Roadmap (2021), and 2050 Carbon Neutral Scenario (2021) have connectivity and have different levels for each plan.

(Page 4, Line 132)

3.The analysis in the third part of the article is too homogeneous and is only a simple recapitulation of existing policies, lacking an overall evaluation of the relevant policies during the period and the directions that should be focused on in the future, which could lead to the content of policies for the next period.

We are grateful to the reviewer for acknowledging the importance of our study and suggesting valuable comments.

Korea's green growth period is an important period that is the basis of the current carbon neutral era. Twenty-seven green technologies were proposed as major policies of the period. R&D investment amount (Line 162) of 27 major green technologies and R&D investment amount (Line 182) of each technology were analyzed.

4.Section 4.1 is titled CCUS policies, but this section is mostly irrelevant, only stating that CCUS has been proposed by the government as a core technology, and lacks specific policies that address CCUS technology, as well as the level of government attention and trends in CCUS technology.

We are grateful to the reviewer for acknowledging the importance of our study and suggesting valuable comments.

Major policies and R&D investment plans by government and ministries in Korea were presented (Line 188). It was reflected in Page 8 and 9. The CCU technology innovation roadmap was analyzed in WBS format (Line 230).

5.Section 4.2 is a simple recapitulation of existing data and is too homogeneous in its analysis.

We are grateful to the reviewer for acknowledging the importance of our study and suggesting valuable comments.

The data from pages 12-14 show the volatility of CCUS' R&D investment in the last 10 years in Korea, and Figure 6 shows that CCS's investment in capture and storage decreases and CCU investment increases, indicating a downward trend in overall CCUS investment. Tables 2 and 3 analyzed the correlation through statistical techniques.

6.The conclusion suggests adding a section on relevant policy insights, indicating the shortcomings of existing CCUS policies and the directions on which they should focus in the future.

We are grateful to the reviewer for acknowledging the importance of our study and suggesting valuable comments.

We reflected it in line 343 of the text.

Reviewer 2 Report

The manuscript has been revised carefully and can be published, however, the format of a few figures and Tables needs to be further improved, e.g.,  Figures 6 and 7, Tables 2 and 3.

Author Response

The manuscript has been revised carefully and can be published, however, the format of a few figures and Tables needs to be further improved, e.g.,  Figures 6 and 7, Tables 2 and 3.

We are grateful to the reviewer for acknowledging the importance of our study and suggesting valuable comments.

We reflected the reviewer's opinion in the text.

Line 253, Line 259, Line 276, Line 283
